# Tracking a spin-polarized superconducting bound state across a quantum phase transition

Sujoy Karan [1] ✉, Haonan Huang [1], Alexander Ivanovic[2], Ciprian Padurariu [2], Björn Kubala [2,3], Klaus Kern [1,4], Joachim Ankerhold [2] & Christian R. Ast [1] ✉

The magnetic exchange coupling between magnetic impurities and a superconductor induce so-called Yu-Shiba-Rusinov (YSR) states which undergo a quantum phase transition (QPT) upon increasing the exchange interaction beyond a critical value. While the evolution through the QPT is readily observable, in particular if the YSR state features an electron-hole asymmetry, the concomitant change in the ground state is more difficult to identify. We use ultralow temperature scanning tunneling microscopy to demonstrate how the change in the YSR ground state across the QPT can be directly observed for a spin-1/2 impurity in a magnetic field. The excitation spectrum changes from featuring two peaks in the doublet (free spin) state to four peaks in the singlet (screened spin) ground state. We also identify a transition regime, where the YSR excitation energy is smaller than the Zeeman energy. We thus demonstrate a straightforward way for unambiguously identifying the ground state of a spin-1/2 YSR state.

Unpaired spins in impurities coupled to a superconductor induce discrete sub-gap excitations, the Yu-Shiba-Rusinov (YSR) states[1–4], through an exchange interaction produced locally via impurity-superconductor coupling. If the exchange coupling increases beyond a critical value the YSR states undergo a quantum phase transition (QPT) such that the initially free spin becomes screened[5,6]. The transition through a QPT has been attributed to a reversal in the asymmetry of the spectral weight of electron and hole excitation components, which are readily observed in a scanning tunneling microscope (STM)[7–13]. This reversal in spectral weight holds, however, only in the simplest approximation that all higher order effects are ignored. The spectral weight does not reflect the particle-hole asymmetry if, for example, the system is already in the resonant Andreev reflection regime[14] or tunneling paths are interfering[15]. Most crucially, it is a priori not possible with the STM to identify to which side of the quantum phase transition the system belongs.

A straightforward albeit indirect and not entirely unambiguous way to manipulate the ground state of an atomic scale YSR resonance is to change the impurity-substrate coupling if the YSR impurity is susceptible to the atomic forces acting between tip and sample in the STM tunnel junction[7,11,16–18]. The ambiguity arises because it is not a priori clear whether the impurity-substrate coupling increases or decreases upon reducing the tip-sample distance. This calls for an unambiguous manifestation going beyond auxiliary measurements[18] to distinguish the ground state of the YSR excitation.

An independent observation identifying the ground state of the system across the QPT can be made by placing the YSR state in a Josephson junction (0-$\pi$ transition)[19]. Also, the zero-field splitting of YSR excitations due to effective anisotropic interactions in high-spin systems has been used to assign the ground state of different molecules on either side of the QPT[20]. While different YSR states have been studied with the STM in the presence of a magnetic field[21–24], a

[1]Max Planck Institute for Solid State Research, Heisenbergstraße 1, 70569 Stuttgart, Germany. [2]Institute for Complex Quantum Systems and IQST, Universität Ulm, Albert-Einstein-Allee 11, 89069 Ulm, Germany. [3]Institute for Quantum Technologies, German Aerospace Center (DLR), Wilhelm-Runge-Straße 10, 89081 Ulm, Germany. [4]Institut de Physique, Ecole Polytechnique Fédérale de Lausanne, 1015 Lausanne, Switzerland. ✉e-mail: s.karan@fkf.mpg.de; c.ast@fkf.mpg.de

continuous evolution of the YSR state across the QPT in a magnetic field has not been observed as it has in mesoscopic systems such as superconducting quantum dots[25]. Despite the different length scales between the mesoscopic systems and the STM tunnel junction, the YSR modeling is remarkably similar, when the size of the system (e.g., quantum dot) is smaller than the superconducting coherence length. The challenge in observing a sizeable Zeeman splitting in a YSR state lies with the typically rather small critical magnetic field that quenches superconductivity. Here, we circumvent this problem by placing the YSR state at the tip apex[19,23,24,26], where the superconductor is dimensionally confined, such that the critical field is considerably enhanced (Meservey-Tedrow-Fulde (MTF) effect)[27–29]. We use an ultralow temperature STM at 10 mK to reduce the thermal energy much below the Zeeman energy and trace the spectral signatures associated with the changes in the YSR ground state across the QPT by continuously changing the impurity-substrate coupling (see Fig. 1a).

## Results

A typical spectrum measured with a YSR functionalized superconducting vanadium tip on a superconducting V(100) sample at 10 mK is shown in Fig. 1b. The electron and hole parts of the YSR state with energy $\varepsilon$ appear at a bias voltage $eV = \pm(\varepsilon + \Delta_s)$ as prominent peaks. Due to the superconducting sample, the YSR peaks shift by the sample gap $eV = \pm\Delta_s$ away from zero bias voltage. The coherence peaks at $eV = \pm(\Delta_t + \Delta_s)$, which is the sum of the tip and sample gaps, are small indicating a dominant transport channel through the YSR state. We change the impurity-substrate coupling by varying the tip-sample distance, which modifies the atomic force acting on the impurity[30,31]. This concomitantly changes the exchange coupling $J$ causing an evolution of the YSR energy $\varepsilon$ as shown in Fig. 1c. At a critical exchange coupling $J_T$, when the YSR energy is at zero, the system moves across a QPT such that the free impurity-spin ($J < J_T$) becomes screened ($J > J_T$) bringing about a change in the fermionic parity of the ground state[32]. This scenario is schematically depicted in the insets of Fig. 1c, where a doublet ($S = 1/2$) transforms into a singlet ($S = 0$) leading to the screening of the impurity spin.

Figure 1d shows how the YSR peaks evolve with the junction transmission $\tau = G_N/G_0$ ($G_N$: normal state conductance; $G_0 = 2e^2/h$: conductance quantum with $e$ being the elementary charge and $h$ Planck's constant). The YSR peaks evolve continuously reaching the bias voltage closest to zero at the QPT. Because of the shift of the YSR state by the superconducting gap $\Delta_s$ of the other electrode (the substrate in this case) in the conductance spectrum, the zero crossing at the QPT is not observed directly. An inversion of the asymmetry in the YSR peak intensities is clearly visible, when the electron and hole excitation components switch sides across the QPT. However, it is not possible to judge from the tunneling spectra alone, on which side of the QPT the system is.

Turning on a magnetic field, the $S = 1/2$-state splits into two levels, which is discussed in Fig. 2. In the free spin regime, the spin down state (see Fig. 3a) turns into the non-degenerate ground state. Its higher lying spin-flipped partner is thermally not populated due to the extremely low temperature of 10 mK. Only the screened $S = 0$-state appears as a transport channel lying energetically above the doublet. Since it does not change in the magnetic field, it induces only one spectral feature on either side of the Fermi level. In contrast, beyond the QPT, the $S = 0$-state becomes the ground state and charge transfer is possible through the spin-doublet (details see below). This can be seen in Fig. 2a, which shows two representative differential conductance spectra on either side of the QPT at a magnetic field of 750 mT. The orange spectrum shows two features (one on either side of the Fermi level), which indicates that the system is in the free spin regime. The sample is already normal conducting at 750 mT, such that there is no shift of the YSR peak by $\Delta_s$. The YSR tip is still superconducting due to the MTF effect. In the screened spin regime, ground state and excited state are interchanged, such that now two transitions into the upper and lower Zeeman split $S = 1/2$ levels are possible from the single ground state $S = 0$ level. As a result, the spectrum measured in the screened spin regime (the blue curve) shows four spectral features (two on either side of the Fermi level). This distinction is only possible, if the Zeeman energy is much larger than the thermal energy. If this is not the case, two spectral features will be visible on either side of the

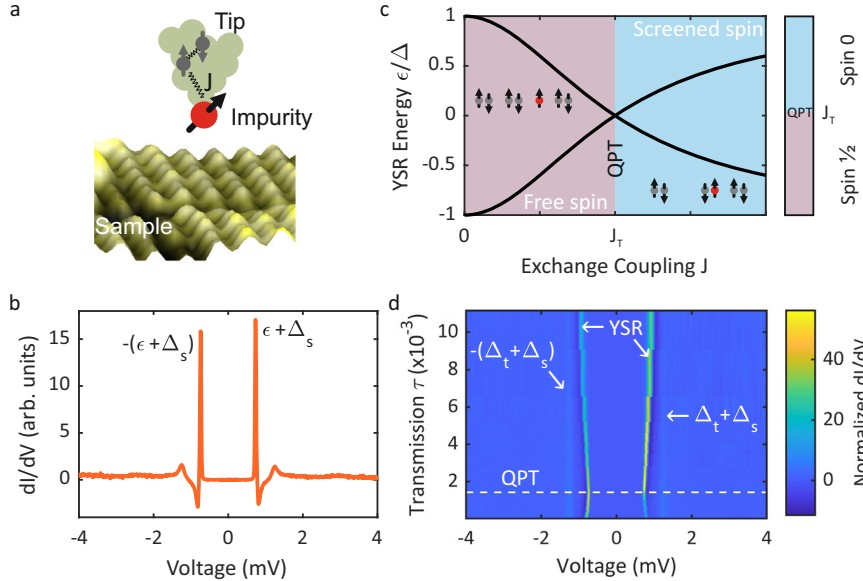

**Fig. 1 | Yu-Shiba-Rusinov (YSR) state in the vicinity of a QPT. a** Schematic of the tunnel junction incorporating a magnetic impurity at the tip apex. **b** Differential conductance spectrum at zero field showing the impurity-induced YSR states at $eV = \pm(\varepsilon + \Delta_s)$. The spectrum was recorded at $\tau = 1.54 \times 10^{-3}$ with the feedback opened at 4 mV. **c** The YSR excitation energy $\varepsilon$ vs. magnetic exchange coupling $J$. At the crossing of the YSR energies, the system undergoes a quantum phase transition (QPT) from a free spin doublet into a screened spin singlet state (see inset). **d** Normalized differential conductance spectra as function of junction transmission $\tau$. The QPT occurs, when the YSR peaks are closest to zero. The YSR peak crossing is not directly visible because both tip and sample are superconducting shifting the YSR peaks by the sample gap $\pm\Delta_s$. **b, d** The coherence peaks are visible at the sum of the tip and sample gap $eV = \pm(\Delta_t + \Delta_s)$.

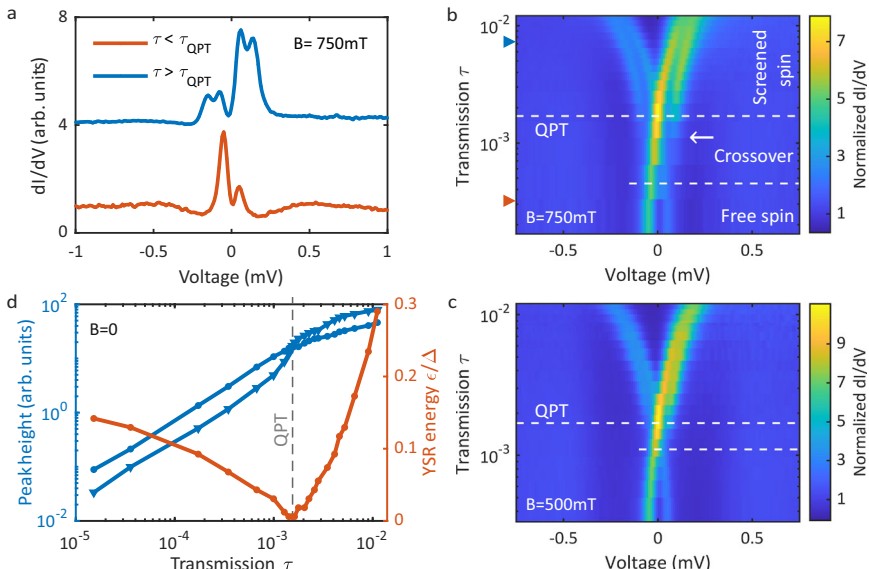

**Fig. 2 | Magnetic field dependence of YSR states across quantum phase transition (QPT). a** Differential conductance spectra at a magnetic field of $B = 750$ mT at two junction transmissions, one below and one above the QPT marked by arrows in (**b**). **b** Differential conductance map at 750 mT as function of junction transmission revealing the shift of the YSR state across the QPT. The

sample is normal conducting, so that the spectral features cross at the Fermi level. **c** Same as (**b**) for 500 mT with a correspondingly reduced Zeeman splitting. **d** YSR peak heights (blue) at zero field of the left and right peak. The height inverts across the QPT, where the YSR energies $\varepsilon$ (orange) are zero. The error bars are within the size of the markers.

QPT and a more detailed analysis of the spectral weight has to be done to distinguish the ground states[20].

As has been demonstrated before[7–13,18,19,26], we exploit the changing atomic forces in the tunnel junction when reducing the tip-sample distance to change the impurity-superconductor coupling thereby shifting the YSR state energy. We note that depending on the particular system, the impurity-substrate coupling can increase or decrease during tip approach (see Supplementary Note 1). The evolution of the YSR state through the QPT for two different magnetic fields are shown in Fig. 2b, c as function of the tunnel junction transmission $\tau$ (i.e., junction conductance). Here, we can see directly that the screened spin regime featuring four spectral peaks is at higher transmissions and the free spin regime featuring only two spectral peak is at lower transmissions. This actually implies that the impurity-superconductor coupling increases with increasing transmission, which is verified by an additional analysis of the Kondo effect at higher magnetic fields below. The data in Fig. 2b was taken at 750 mT, which results in a stronger Zeeman splitting than the data in Fig. 2c, which was taken at 500 mT having less Zeeman splitting. Still, both data sets show qualitatively the same behavior across the QPT as expected from the discussion above.

In addition to these two regimes, we found a crossover regime, where the two outer spectral features extend into the free spin regime, which is seen for both magnetic field values in Fig. 2b, c. Due to the higher magnetic field in Fig. 2b than in c, the crossover regime is also wider. The crossover regime marks a small region, where the excitation energy of the YSR state is smaller than the Zeeman splitting ($\varepsilon < E_Z$). The outer spectral feature (marked by the arrow in Fig. 2b) in the crossover regime is a combination of quasiparticle tunneling from the thermally excited YSR state, which becomes exponentially suppressed as the YSR energy increases, and two-electron tunneling processes, i.e., resonant Andreev processes (see below and Supplementary Note 2).

The different regimes for a spin-1/2 impurity are schematically displayed in Fig. 3a. The screened spin regime (blue shade), where the exchange coupling is strong $J > J_T$, features an $S = 0$ ground state and a Zeeman split excited $S = 1/2$ state. Two transitions are possible ($|0\rangle \rightarrow |\downarrow\rangle$ and $|0\rangle \rightarrow |\uparrow\rangle$) as shown on the right blue panel. Lowering the exchange coupling, the $|\downarrow\rangle$ state becomes the ground state at the

QPT ($J = J_T$). Interestingly, in the crossover regime the excited state $|0\rangle$ is energetically between the ground state $|\downarrow\rangle$ and the Zeeman split state $|\uparrow\rangle$. Therefore, both thermally excited tunneling and two-electron tunneling processes are possible resulting in the outer spectral feature (white arrow in Fig. 2b). As a consequence, two transitions can be observed ($|\downarrow\rangle \rightarrow |0\rangle$ and $|0\rangle \rightarrow |\uparrow\rangle$). Further reducing the exchange coupling into the free spin regime reduces the visible transitions to one ($|\downarrow\rangle \rightarrow |0\rangle$), because the Zeeman split state $|\uparrow\rangle$ cannot be thermally excited at 10 mK. We can reproduce the experimental findings theoretically by calculating a tunneling current from a master equation involving both single electron and two electron processes (for details see Supplementary Note 2). The calculation in Fig. 3b has been done for a magnetic field of 750 mT comparable to the experimental data in Fig. 2b. All the features that we observed experimentally are reproduced in the calculations.

In order to independently verify the evolution of the YSR state through the QPT, we take a closer look at the YSR peak height and the resulting Kondo effect in the normal conducting state. The evolution of the YSR peak height is plotted in Fig. 2d in blue for the left and right peak as function of junction transmission. In the same graph the YSR peak energy is shown in orange. At the QPT (vertical dashed line), the YSR energy is zero and the peak height reverses indicating the QPT. This reversal is observable so clearly because resonant Andreev processes have not yet become significant. Further, we increase the magnetic field to 2.75 T such that both tip and sample become normal conducting and a Kondo peak appears[33–36]. This is shown in Fig. 4a, where the Kondo peak around zero bias voltage is displayed as a function of the junction transmission $\tau$. We already see that the splitting of the Kondo peak in the magnetic field decreases as the transmission increases, which indicates that the Kondo temperature increases with increasing transmission. The higher Kondo temperature implies a stronger screening, which means that the Kondo peak starts splitting at a higher critical magnetic field $B_c$. We have fitted the Kondo spectra using numerical renormalization group (NRG) theory[37]. This allows us to directly determine the Kondo temperature $T_K$ from the microscopic parameters extracted from the fit. The extracted Kondo temperature is shown in Fig. 4b as the blue line. It monotonously

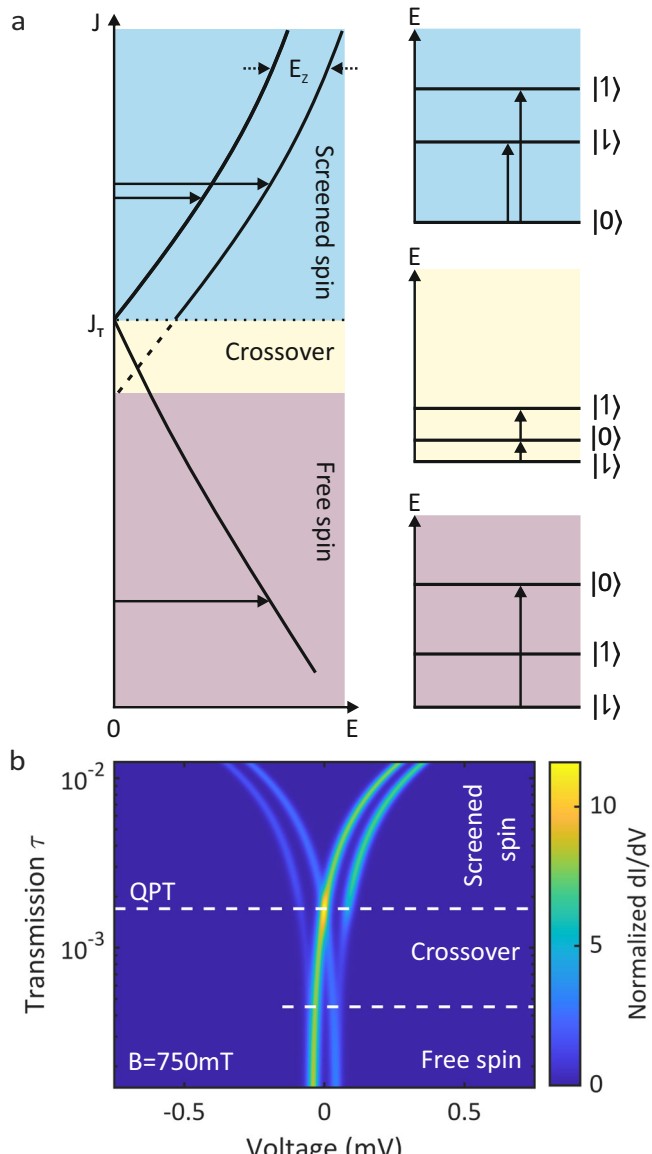

**Fig. 3 | Theoretical modeling of the Zeeman splitting across the quantum phase transition (QPT). a** The sketches illustrate the level structures of a YSR state at finite magnetic field in different regimes across the QPT. The Zeeman effect lifts the spin degeneracy of the doublet state leading to two possible transitions in the screened spin regime (shaded blue). The YSR states are split by the Zeeman energy $E_z = g\mu_B B$, where $g$ is the $g$-factor and $\mu_B$ is the Bohr magneton. The crossover regime (shaded yellow) in the proximity of the QPT still allows for a second transition involving thermal excitation of the YSR state and two-electron tunneling processes. The free spin regime (shaded brown) allows only for one transition. **b** Calculation of the differential conductance spectra across the QPT based on a master equation including single electron and two electron processes. The experimental data in Fig. 2b is well reproduced.

increases with increasing junction transmission $\tau$, which corroborates the previous finding that the exchange coupling increases with increasing transmission (cf. Fig. 2b, c)[18,38]. We also extracted the critical field $B_c$, where the Kondo peaks starts splitting, as a function of transmission. The values for the critical field $B_c$ are plotted in Fig. 4b as an orange line. The critical field increases with increasing transmission and follows the Kondo temperature very well. This corroborates very well the increase in impurity-substrate coupling for an increasing junction transmission. We further find a relation of $k_B T_K = \alpha \mu_B B_c$ between the Kondo temperature and the critical field with $\alpha = 1.6$, which compares well with what has been found in the literature[39–41].

Furthermore, scaling the Kondo temperature $T_K$ and the YSR energy $\varepsilon$ to the superconducting gap $\Delta$, we compare the evolution across the QPT to the universal behavior predicted by NRG theory[42–44]. The blue data points in Fig. 4c show the evolution of the YSR state across the QPT as function of the scaled Kondo temperature, which follows the predicted universal scaling (dashed line) with a slight off-set. This deviation of the data from the universal curve is presumably due to subtle changes in the impurity-substrate coupling as a result of modifications in the atomic forces acting in the junction with and without the applied magnetic field. We, therefore, find a consistent picture for the behavior of the YSR state in a magnetic field across the quantum phase transition.

The evolution of the YSR state splitting across the QPT clearly demonstrates the change in the YSR ground state. For a spin-1/2 system, the nature of the ground state can be straightforwardly identified simply by the number of peaks in the spectrum. For higher order spins, the situation remains simple as long as the system can be assumed to be magnetically isotropic[24]. If the system experiences a magnetic anisotropy, the analysis of the YSR states becomes more cumbersome[21,45]. Still, the evolution in a magnetic field as well as with changing impurity-superconductor coupling (if susceptible to the atomic forces of the tip) greatly facilitates the identification of the ground state maybe even the spin state itself. Another interesting application could be to determine the quasiparticle temperature far below the STM energy resolution as outlined in Supplementary Note 3.

In summary, we present the evolution of a spin-1/2 impurity derived YSR state in a magnetic field across the QPT. We find generally good agreement with mesoscopic measurements across different length scales[25]. Due to the extremely low temperature of the STM, the change from a single feature spectrum (free spin regime) to a double feature spectrum (screened spin regime) is clearly visible. This allows for an unambiguous determination of the ground state of the YSR state.

## Methods

The V(100) single crystal was sputtered (with Ar⁺), annealed to about 925 K, and cooled to ambient temperature repeatedly in ultra-high vacuum, ensuring an atomically flat sample surface. Typical surface reconstructions form with oxygen diffused from the bulk[26,46,47]. A small fraction of surface defects exhibit YSR states[26]. Similarly, we produce YSR states at the vanadium tip apex by repeatedly dipping the tip in situ into the substrate[19,26], which is verified by the conductance spectrum. This gives us full control to reproducibly design and define the junctions under investigation. We choose to use YSR functionalized tips for our experiments as they offered the flexibility to single out those fulfilling the required response to tip approach. Moreover, the YSR tips feature a range of YSR state energies and show a better junction stability at higher conductance than YSR states in the sample.

The experiments were performed in a low-temperature scanning tunneling microscope operating at 10 mK. Differential tunneling conductance ($dI/dV$) spectra were recorded using an open feedback loop with a standard lock-in technique (10 $\mu V_{rms}$, 727.8 Hz). A modulation amplitude of 25 $\mu V_{rms}$ was used for the spectra recorded in the magnetic field. The tunneling current was measured through the tip with the voltage bias applied to the sample.

The calculations for the current based on the master equation are detailed in Supplementary Note 2. To model the Kondo spectra in a magnetic field, we used the numerical renormalization group (NRG) theory in the framework of the single impurity Anderson model (SIAM) as implemented in "NRG Ljubljana" code[48]. We fixed the Hubbard term $U = 10$ to be much larger than the half bandwidth $D = 1$ and modeled the asymmetry of the Kondo spectra using the intrinsic asymmetry parameter $\delta = \epsilon + U/2$ where $\epsilon$ is the impurity level[37]. The best agreement with the experiment corresponds to $\delta = -2$. The only free parameter left for fitting the Kondo spectra is

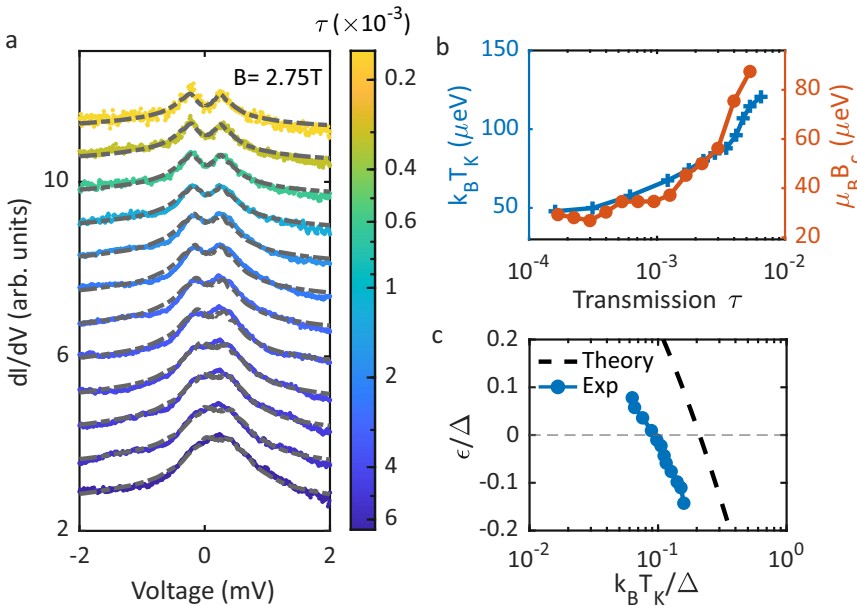

**Fig. 4 | Kondo effect and numerical renormalization group (NRG) analysis.**
**a** Differential conductance spectra featuring a Kondo peak at different junction transmissions $\tau$. Dashed gray lines represent fits of the data to NRG calculations[48]. **b** The Kondo temperature extracted from the fit in (**a**) as function of transmission $\tau$

in blue along with the critical field $B_c$, above which the Kondo peak starts splitting. **c** YSR state energy vs. Kondo temperature both scaled to the superconducting gap $\Delta$. The dashed black curve represents the universal scaling predicted from the NRG model. The error bars are within the size of the markers.

the impurity-substrate coupling $\Gamma$. The Kondo temperature was extracted from the fit through its definition with respect to the SIAM parameters[43,49,50] $k_B T_K = D_{eff} \sqrt{\rho J} \exp(-\frac{1}{\rho J})$ with $\rho J = \frac{8\Gamma}{\pi U} \frac{1}{1-4(\delta/U)^2}$, in which the effective bandwidth satisfies $D_{eff} = 0.182 U \sqrt{1-4(\delta/U)^2}$ for $U \ll 1$ and $D_{eff}$ is a constant for $U \gg 1$[37,43,51].

## Data availability

All data needed to evaluate the conclusions are present in this paper and/or the Supplementary Information. In addition, the data related to this paper are available from the EDMOND Database[52].

## Code availability

The code used for the calculations presented here is available from the authors upon request.

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

## Acknowledgements
The authors thank Carlos Cuevas and Andrea Hofmann for fruitful discussions. This study was funded in part by the ERC Consolidator Grant AbsoluteSpin (Grant No. 681164). J.A. and C.P. gratefully acknowledge financial support by the IQST and the BMBF through QSens (project QComp).

## Author contributions
S.K. did the experiments with support from H.H., K.K., and C.R.A. A.I., C.P., B.K., and J.A. provided theory support. S.K., H.H., and C.R.A. analyzed the data and discussed the results with all authors. S.K. and C.R.A. wrote the manuscript with input from all authors.

## Funding

## Competing interests
The authors declare no competing interests.
