## [Peer Review File · Nature Communications]

Tracking a Spin-Polarized Superconducting Bound State across a Quantum Phase TransitionREVIEWER COMMENTS

Reviewer #1 (Remarks to the Author):

The authors have investigated the tunneling spectra of a junction consisting of a vanadium substrate and a vanadium tip with an $S=1/2$ impurity near the tip apex. The impurity creates the YSR states in the spectrum that exhibits intricate behaviors as functions of a junction transmission and a magnetic field. The junction-transmission dependence indicates a signature of the quantum phase transition of the many-body ground state, which is clearly evidenced by two distinct magnetic-field responses. The authors have performed comprehensive theoretical analyses that quantitatively explain complicated behaviors in the crossover regime. I think that this work provides valuable information in YSR physics, and the paper can be published in Nature Communications after considering the following two comments.

1. There is a similar crossover experiment on a quantum dot in magnetic fields [25]. I understand that it is much harder to implement the experiment in STM than in quantum dots. It would be meaningful if the authors argue the new insight into the basic YSR physics or a potential application suggested from the STM work. At least, the authors should discuss the common and different aspects between the current results and the results of Ref. [25].

2. The nature of the impurity should be discussed. Why does the bond between the impurity and the tip change significantly as a function of transmission? Is it possible to prepare a functionalized YSR tip in the strongly screened regime or in the almost unscreened regime, being robust against the junction transmission?

Reviewer #2 (Remarks to the Author):

The authors measured the tunneling spectrum at 10 mK with a V-tip with a randomly attached impurity at the apex on the vanadium superconductor. They observed the Yu Shiba-Rusinov (YSR) states on the spectrum. By varying the distance between the tip and the superconducting substrate, they observed the evolution of the YSR states and explained their data based on a picture involved with a quantum phase transition (QPT) upon increasing the exchange interaction beyond a critical value. It seems that the data can be described by the model or picture, even in a pedagogical way. However, I don't see too much novelty revealed by the data and explanations, and the story is

interesting only for specialists, in this sense, I would not recommend the publication of this work in Nature Communications which is designed for broader interests.

In addition, before the publication of this work in some way, I have two concerns which may need to be amended.

(1) How can the author achieve a spin-1/2 magnetic impurity by “repeatedly dipping the tip in situ into the substrate”. This is somehow difficult to control the magnetic moment and its orientation.

(2) The junction is actually a SIS junction, thus, theoretical circumstance becomes a bit sophisticated. How the authors distinguish the single particle tunneling or Josephson tunneling in this junction.

Reviewer #3 (Remarks to the Author):

Summary:

In their work "Tracking a Spin-Polarized Superconducting Bound State across a Quantum Phase Transition" the authors use a 10mK STM to study Yu-Shiba-Rusinov states under an applied external magnetic field and by tuning the tip-sample tunneling conductance so that the system is controllably driven through a quantum phase transition between a spin singlet and a spin doublet ground state. They demonstrate on their system, i.e. a superconducting Vanadium tip functionalized by a magnetic impurity (oxygen defect) on a superconducting V(100) surface, that the observation or absence of a Zeeman splitting of the YSR peaks is a direct indication for the actual ground state. In an unprecedented experiment they show the continuous evolution through the QPT where four peaks become two and vice versa. They thereby experimentally resolve a crossover region where the energy of the spin degenerate singlet state lies in between the two levels of the spin split doublet state.

They further support and verify their interpretation of the results with a theoretical model simulation of the QPT crossing and by measuring the evolution of the Kondo temperature through fitting an observed Kondo resonance. Both are in very good agreement to the results and show a consistent picture.

Soundness of the work:

The work is done experimentally very excellent. All relevant results are clearly seen in the presented data.

Significance:

In general, I strongly advocate the significance of carefully done investigations like in this work, using the ultralow temperature to identify separate tunneling processes that would otherwise go unnoticed in order to understand the fundamental physics which might in turn facilitate the development of new experimental methods.

However, in this particular case the article in my opinion strongly resembles another recently published article (Nature Physics 18, 893 (2022).) by the same authors both in the used system under investigation (functionalized V tip on V(100)) and in the method of continuously driving a QPT through tuning tunneling conductance by changing tip-sample distance, as well as supporting this by extracting the Kondo temperature.

Further, the claim "We thus provide a straightforward way for unambiguously identifying the ground state of a spin-1/2 YSR state." should be considered to be reduced to "We demonstrate..." since this was already shown before by Machida et al. (<https://doi.org/10.1103/PhysRevResearch.4.033182>) who used the equivalent way of applying a magnetic field on a YSR functionalized tip to observe the Zeeman splitting without quenching superconductivity.

Both above mentioned articles are of course correctly cited in this work, yet they finally leave the observation of the crossover region when the system is tuned through the QPT in an applied field as the only essential novelty in this article which in my opinion limits its significance.

More detailed comments and questions:

- "The challenge in observing a sizeable Zeeman splitting in a YSR state" is stated to be the quenched superconductivity in small applied fields. In my opinion the bigger limitation is the lowest reachable temperature and thus energy resolution needed to observe this feature. The method of using the spatial confinement in the tip to maintain superconductivity was also used in other works that could be cited here (e.g. <https://www.science.org/doi/10.1126/sciadv.abd7302> , <https://doi.org/10.1103/PhysRevResearch.4.033182>).

- In Fig. 1b: I miss measurement parameters for reference: transmission, opening bias voltage.

- Tip dependence: It is mentioned that impurity-substrate coupling can increase or decrease during tip approach depending on the particular system. Does that mean that already a slightly changed tip configuration in the same material system (V tip on V(100)) can flip the trend? It seems like this is the case since the authors observed the opposite trend (decreased coupling when the tip is approached) in a previous work with the same experimental setup. Is there any data of this experiment in applied

field with a tip that has the opposite behavior? It might be a nice supplementary data to demonstrate an even more robust picture.

- Zeeman shift: In the free spin regime only one spectral feature on either side of the Fermi level is observed. Those features should shift in energy depending on the applied field. Is there any field dependence data that shows this?

- Magnetic field dependence: Would it be possible to get a quantitative connection between the transmission τ and the exchange coupling J by measuring the size of the crossover region in dependence of the magnetic field?

- Fig. 2: caption of b: I think "dispersion" is not correct here since in solid state physics dispersion is a change with crystal momentum k . Simply putting "shift" might be better.

- Fig. 2b,c: The logarithmic scale leads to the impression that there are less points taken before and at the QPT, especially in the crossover region, where it is presumably the most interesting to look at. Nevertheless, the presented data is clear and sufficient in my opinion, so this is just a comment.

- Fig. 2d: Although it might destroy the used color codes, I think the two blue lines should be more distinguishable than just by their point mark, especially in the region where they cross. Maybe a little different nuance of blue would improve the visibility.

Overall, I think this is a very sound work which stands out in the field simply by its quality and clearness. Novel experimental observations are presented and provided with a robust explanation and theoretical modelling. However, I would evaluate their significance regarding progress in the field as medium for the above stated reasons. Nevertheless, I think the article meets all requirements to be published in Nature Communications.

Reply to Referee's Comments

We thank the reviewers for evaluating our manuscript. Below, we outline the Referee's comments in detail. Note that a sentence in the abstract is rephrased to adapt to the word limit. Changes in the manuscript are marked in blue.

Answers to Referee #1:

The authors have investigated the tunneling spectra of a junction consisting of a vanadium substrate and a vanadium tip with an $S=1/2$ impurity near the tip apex. The impurity creates the YSR states in the spectrum that exhibits intricate behaviors as functions of a junction transmission and a magnetic field. The junction-transmission dependence indicates a signature of the quantum phase transition of the many-body ground state, which is clearly evidenced by two distinct magnetic-field responses. The authors have performed comprehensive theoretical analyses that quantitatively explain complicated behaviors in the crossover regime. I think that this work provides valuable information in YSR physics, and the paper can be published in Nature Communications after considering the following two comments.

We thank the Reviewer for their positive assessment of our work and recommendation for publication. We appreciate their recognition of the work being informative and valuable to YSR physics. Their comments are addressed in detail below.

1. There is a similar crossover experiment on a quantum dot in magnetic fields [25]. I understand that it is much harder to implement the experiment in STM than in quantum dots. It would be meaningful if the authors argue the new insight into the basic YSR physics or a potential application suggested from the STM work. At least, the authors should discuss the common and different aspects between the current results and the results of Ref. [25].

While both our work and the work presented in Ref. [25] show the evolution of a superconducting bound state in presence of a magnetic field, we operate in an entirely different length scale in contrast to the mesoscopic quantum dot community, as already pointed out by the Referee. Additionally, an STM junction is intrinsically highly asymmetrically coupled and the exchange coupling can be controlled without the need of a gated electrode. However, the most significant advance is the observation of a hitherto unobserved crossover regime near the QPT, where the thermal excitation leads to the splitting of the YSR state much like what we see for a screened spin state, although the system ground state corresponds to the free spin regime. Thus, merely seeing a YSR state split into two levels, when close to the QPT, does not necessarily confirm the screened spin regime. Therefore, our results also present means to identify, on which side of the QPT a YSR state is, which is extremely difficult by just looking at the quasiparticle spectrum at zero magnetic field. In addition, because the crossover regime is enabled

FIG. R1: **Non moving YSR state.** Differential conductance map as function of junction transmission. No shift in energy of the YSR state is observed. The zero-bias Josephson peak is also visible at higher junction transmission.

by quasiparticle excitations that require local quasiparticle parity breaking at the position of the impurity, our experiment is sensitive to the effective temperature of local quasiparticle excitations in the tip. We should in principle be able to quantify the local quasiparticle temperature from our experiment.

2. The nature of the impurity should be discussed. Why does the bond between the impurity and the tip change significantly as a function of transmission? Is it possible to prepare a functionalized YSR tip in the strongly screened regime or in the almost unscreened regime, being robust against the junction transmission?

As has been shown before, the atomic force acting in a junction at low transmission, where we operate, is weakly attractive, but not zero (see Ref. [31]). The change in atomic forces due to tip displacement results in atomic relaxation at the tunnel junction which changes the impurity-superconductor coupling by pulling on the impurity (see Ref. [7]), thereby changing its bonding strength to the bulk vanadium. Within the Anderson impurity model, the impurity-substrate coupling has direct influence on the YSR state energy. We note that the structural relaxations could also lead to a change in the local density of states, such that the effective impurity-substrate coupling may actually increase upon approaching the tip, as has been observed in Ref. [11] and [18].

For this experiment, we were looking for YSR tips that show the required response to the tip approach. However, there are cases where the impurity is more rigidly bound to the tip and, hence, is not susceptible to the atomic forces acting in the junction, as pointed out by the referee. In these cases, the YSR energy does not change with the experimentally accessible junction transmission. An example is shown in Fig. R1. We have now added a section in the supplementary material to discuss more on this point.

Answers to Referee #2:

The authors measured the tunneling spectrum at 10 mK with a V-tip with a randomly attached impurity at the apex on the vanadium superconductor. They observed the Yu Shiba-Rusinov (YSR) states on the spectrum. By varying the distance between the tip and the superconducting substrate, they observed the evolution of the YSR states and explained their data based on a picture involved with a quantum phase transition (QPT) upon increasing the exchange interaction beyond a critical value. It seems that the data can be described by the model or picture, even in a pedagogical way. However, I don't see too much novelty revealed by the data and explanations, and the story is interesting only for specialists, in this sense, I would not recommend the publication of this work in Nature Communications which is designed for broader interests.

In addition, before the publication of this work in some way, I have two concerns which may need to be amended.

We thank the Referee for the assessment of our work. While the Referee does not question the scientific validity of our work, they decline publication citing our work as too specialized to appear in Nature Communications in disagreement with the two other reviewers' opinions. A couple of concerns have been raised by the Referee resulting from a misunderstanding, which we are happy to clarify below.

(1) How can the author achieve a spin-1/2 magnetic impurity by "repeatedly dipping the tip in situ into the substrate". This is somehow difficult to control the magnetic moment and its orientation.

The preparation of a functionalized tip with a spin-1/2 impurity is straightforward and based on an automated trial-and-error process. We have reproduced functionalized YSR tips many times. During the time we have worked with these impurities, we have collected multiple indications for their spin-1/2 character, such as only one YSR state in the gap (i.e. two peaks in the spectrum), Kondo peak splitting beyond a critical magnetic field, and transport through two YSR states. A number of these results with impurities at the vanadium tip as well as in the vanadium sample are already published (see Ref. [26], [19]; Huang et al., Phys. Rev. Res. 3, L032007 (2021); Huang et al., arXiv:2212.11332), all of which indicate that the impurities in vanadium are spin-1/2 systems giving an overall consistent picture. Since there is no anisotropy for a spin-1/2 impurity, it will align along the applied magnetic field which is normal to the sample surface in our case.

(2) The junction is actually a SIS junction, thus, theoretical circumstance becomes a bit sophisticated. How the authors distinguish the single particle tunneling or Josephson tunneling in this junction.

This point is not a concern, but simply a misunderstanding. The evolution of the YSR ground state across the quantum phase transition (QPT) is measured in a magnetic field at which the sample is not superconducting, but only the tip remains superconducting due to the Meservey-Tedrow-Fulde (MTF) effect. Hence, we have a SIN junction, not a SIS junction, which precludes Josephson tunneling.

Answers to Referee #3:

Summary:

In their work "Tracking a Spin-Polarized Superconducting Bound State across a Quantum Phase Transition" the authors use a 10mK STM to study Yu-Shiba-Rusinov states under an applied external magnetic field and by tuning the tip-sample tunneling conductance so that the system is controllably driven through a quantum phase transition between a spin singlet and a spin doublet ground state. They demonstrate on their system, i.e. a superconducting Vanadium tip functionalized by a magnetic impurity (oxygen defect) on a superconducting V(100) surface, that the observation or absence of a Zeeman splitting of the YSR peaks is a direct indication for the actual ground state. In an unprecedented experiment they show the continuous evolution through the QPT where four peaks become two and vice versa. They thereby experimentally resolve a crossover region where the energy of the spin degenerate singlet state lies in between the two levels of the spin split doublet state. They further support and verify their interpretation of the results with a theoretical model simulation of the QPT crossing and by measuring the evolution of the Kondo temperature through fitting an observed Kondo resonance. Both are in very good agreement to the results and show a consistent picture.

Soundness of the work:

The work is done experimentally very excellent. All relevant results are clearly seen in the presented data.

Significance:

In general, I strongly advocate the significance of carefully done investigations like in this work, using the ultralow temperature to identify separate tunneling processes that would otherwise go unnoticed in order to understand the fundamental physics which might in turn facilitate the development of new experimental methods.

We greatly appreciate the Referee's positive assessment of our work and thank the referee for the recommendation for publication. We are happy to answer all their questions below.

However, in this particular case the article in my opinion strongly resembles another recently published article (Nature Physics 18, 893 (2022).) by the same authors both in the used system under investigation (functionalized V tip on V(100)) and in the method of continuously driving a QPT through tuning tunneling conductance by changing tip-sample distance, as well as supporting this by extracting the Kondo temperature.

We would like to point out that except for the continuously going through the QPT and supporting this evolution by extracting the Kondo temperature, which are both just means to support our conclusions, the main focus of these two papers is quite different ($0-\pi$ junction vs. Zeeman splitting). Granted the discussed phenomena are both consequences of the QPT, but the resulting observations manifest themselves in different ways. Specifically, our previous work studies the Josephson current peak, while here,

due to the magnetic field, we have an SIN junction, which shows no Josephson current.

Further, the claim “We thus provide a straightforward way for unambiguously identifying the ground state of a spin-1/2 YSR state.” should be considered to be reduced to “We demonstrate...” since this was already shown before by Machida et al. (<https://doi.org/10.1103/PhysRevResearch.4.033182>) who used the equivalent way of applying a magnetic field on a YSR functionalized tip to observe the Zeeman splitting without quenching superconductivity.

Both above mentioned articles are of course correctly cited in this work, yet they finally leave the observation of the crossover region when the system is tuned through the QPT in an applied field as the only essential novelty in this article which in my opinion limits its significance.

We followed the Referee’s suggestion and reworded the corresponding sentence. Nevertheless, we want to point out here that the paper by Machida *et al.* did not demonstrate the continuous evolution across the QPT. We did demonstrate the continuous evolution across the QPT for one and the same system, which we believe to be an important advancement as it allowed us to identify the crossover regime.

More detailed comments and questions:

- “The challenge in observing a sizeable Zeeman splitting in a YSR state” is stated to be the quenched superconductivity in small applied fields. In my opinion the bigger limitation is the lowest reachable temperature and thus energy resolution needed to observe this feature. The method of using the spatial confinement in the tip to maintain superconductivity was also used in other works that could be cited here (e.g. <https://www.science.org/doi/10.1126/sciadv.abd7302>, <https://doi.org/10.1103/PhysRevResearch.4.033182>).

We thank the Referee for pointing this out. We have added the suggested citations in the introduction.

- In Fig. 1b: I miss measurement parameters for reference: transmission, opening bias voltage.

We are sorry for not being more informative about the spectrum presented in Fig. 1b. We now added all the information in the figure caption that the Referee asked for.

- Tip dependence: It is mentioned that impurity-substrate coupling can increase or decrease during tip approach depending on the particular system. Does that mean that already a slightly changed tip configuration in the same material system (V tip on V(100)) can flip the trend?

Yes, two differently prepared tips can exhibit different behavior. This does not happen for the same tip unless the tip is moved really close to the sample as in Ref. [7], which we never do.

FIG. R2: **Magnetic field dependence of a YSR state across QPT.** Differential conductance map at 500 mT as function of junction transmission. The system belonging initially to the screened spin regime features a spin-split state at low transmission.

It seems like this is the case since the authors observed the opposite trend (decreased coupling when the tip is approached) in a previous work with the same experimental setup. Is there any data of this experiment in applied field with a tip that has the opposite behavior? It might be a nice supplementary data to demonstrate an even more robust picture.

We also have observed the opposite trend. To accommodate the referee's request, we include a data set (as shown in Fig. R2) in the supplementary material where impurity-substrate coupling decreases with increasing junction transmission.

- Zeeman shift: In the free spin regime only one spectral feature on either side of the Fermi level is observed. Those features should shift in energy depending on the applied field. Is there any field dependence data that shows this?

Indeed the spectral features associated to the YSR state in the free spin regime shift in energy upon changing the magnetic field. However, we think the motion of the peaks in this case will not offer a conclusive information for the following reasons. The shift in energy of a moving Shiba state is driven by the change in atomic forces in the junction which is not only a function of junction transmission but also seems to depend on the strength of the magnetic field, likely leading to a noticeable deviation from the predicted universal scaling behavior (see Huang et al., arXiv:2212.11332).

- Magnetic field dependence: Would it be possible to get a quantitative connection between the transmission τ and the exchange coupling J by measuring the size of the crossover region in dependence of the magnetic field?

To establish a connection between the transmission τ and the exchange coupling J , the formula for the energy of the YSR state can be inverted as has been done in Ref. [18]. We see no direct relation with the width of the crossover regime, which according to the

definition in our manuscript (see Fig. 4) depends only on the size of the Zeeman splitting (i.e. on the strength of the magnetic field).

- Fig. 2: caption of b: I think "dispersion" is not correct here since in solid state physics dispersion is a change with crystal momentum k . Simply putting "shift" might be better.

We have made the corresponding change.

- Fig. 2b,c: The logarithmic scale leads to the impression that there are less points taken before and at the QPT, especially in the crossover region, where it is presumably the most interesting to look at. Nevertheless, the presented data is clear and sufficient in my opinion, so this is just a comment.

In principle we agree with the referee, but we found that the present form of the figures suited best for visualizing the crossover regime clearly at the center of the frame, as has also been realized by the referee. We, thus, decide to stick to their present display.

- Fig. 2d: Although it might destroy the used color codes, I think the two blue lines should be more distinguishable than just by their point mark, especially in the region where they cross. Maybe a little different nuance of blue would improve the visibility.

We agree with the referee and have optimized the data point markers to increase the visibility of the crossing between two blue lines.

Overall, I think this is a very sound work which stands out in the field simply by its quality and clearness. Novel experimental observations are presented and provided with a robust explanation and theoretical modelling. However, I would evaluate their significance regarding progress in the field as medium for the above stated reasons. Nevertheless, I think the article meets all requirements to be published in Nature Communications.

We greatly appreciate the reviewer's comments in praise of our work and thank them for their recognition of the manuscript for publication.

REVIEWER COMMENTS

Reviewer #1 (Remarks to the Author):

The authors have responded adequately to most of the comments made by me and other reviewers. Regarding my previous comment on the similarities and differences between the current work and the quantum dot experiments, the authors state in the reply that the length scales are different and that it would be possible to quantify the local quasiparticle temperature. These should be discussed in the main text. It is an interesting proposal that the spectrum in the QPT regime can be used to estimate the temperature of the tunneling region. Why is this unique for the STM configuration but not for the quantum dot? Also, the temperature evaluation should be done using the data presented. If this is difficult at the moment, please discuss in the main text what should be done in the future.

I think it is necessary to compare the results of STM and quantum dot in the main text. Even if they are completely the same except for the length scale, the value of the authors' interesting work does not come down. The robustness of the YSR physics is confirmed across platforms regardless of the length scale.

Reviewer #2 (Remarks to the Author):

I reevaluated the manuscript by reading into the paper. Although some interests arose in my second reading, I persist to think that the work is too specialized, not for a general or broad interests. For example, the YSR states have been known for many years, it can help to resolve the pairing symmetry of some new and unknown superconductors. This tip-distance (transmission rate after the author's terming) controlled physics is quite complicated with this novel junction, which can generally give rise to some evolution of the spectrum. I have many concerns about the magnetic moment on the tip apex, the junction itself, and the interpretation of the output results. Thus I still hold the negative attitude to this work.

1. Firstly, the authors believe that the magnetic moment on a vanadium adatom on the very apex is spin $1/2$, this needs to be well proved. I think this is a very difficult point. The exact way for how the vanadium atom is attached at the tip is very crucial for the final results. For example, is it side-attached or exactly on the tip? I believe this can give enormous influence on the final results, including the YSR states. I must say that the original proposed YSR states appear at or nearby the

magnetic impurity embedded in a body superconductor, it is a collective effect and even some Friedel oscillations of bound states can be observed. Now the magnetic impurity is at the tip, that is really something special, the original picture of YSR states is greatly altered.

2. A strange configuration is chosen by the authors here. They used a vanadium tip and a vanadium substrate, both are superconductive at zero field. They wrote in the paper that "The sample is already normal conducting at 750 mT, such that there is no shift of the YSR peak by Δ_s . The YSR tip is still superconducting due to the MTF effect." I wonder why the authors don't use a non-superconductive metal as the substrate, like Cu or Au crystal. In that case, the output would be more convincing and easy to explain.

Reviewer #3 (Remarks to the Author):

I thank the authors for considering my suggestions and for their explanations.

I stay with my opinion that this is sound work which brings advances for the field. Therefore the results should be published.

Reply to Referee's Comments

We thank the reviewers for evaluating our manuscript again. Below, we outline the Referee's comments in detail. Changes in the manuscript are marked in blue.

Answers to Referee #1:

The authors have responded adequately to most of the comments made by me and other reviewers. Regarding my previous comment on the similarities and differences between the current work and the quantum dot experiments, the authors state in the reply that the length scales are different and that it would be possible to quantify the local quasiparticle temperature. These should be discussed in the main text. It is an interesting proposal that the spectrum in the QPT regime can be used to estimate the temperature of the tunneling region. Why is this unique for the STM configuration but not for the quantum dot? Also, the temperature evaluation should be done using the data presented. If this is difficult at the moment, please discuss in the main text what should be done in the future.

I think it is necessary to compare the results of STM and quantum dot in the main text. Even if they are completely the same except for the length scale, the value of the authors' interesting work does not come down. The robustness of the YSR physics is confirmed across platforms regardless of the length scale.

We thank the Reviewer again for recommending publication with their valuable suggestions. We are happy that the Reviewer finds our response adequate and suggests to include the discussion on comparisons between our work and the quantum dot experiments in the main text. We also appreciate the referee's recognition of the value of our work and acknowledging science across different platforms. Following the suggestion of the reviewer, we have now added the discussion on this point in the main text, which is highlighted in blue.

To explain the thermometry idea in some more detail, we envision building on the advantages of the STM setup by realizing thermometry of the quasiparticle temperature at the atomic scale. At 15 mK, it is not possible to measure temperature broadening directly from the YSR conductance peak due to the much larger broadening due to $P(E)$ -type voltage noise that limits the energy resolution. Our current experiment provides a workaround, by exploiting the quantum phase transition. The conductance shows a pronounced peak when the YSR state crosses the phase transition. The peak is defined along the voltage axis (centered at zero voltage), but also along the parameters that control the phase transition: the exchange coupling strength J and the magnetic field B . Integrating the conductance along the voltage results in a peak with a well defined width in either J or B . This width is independent of voltage noise and is a good measure of the electronic temperature.

A similar electronic temperature measurement is possible for quantum dots. However, for quantum dots there are straightforward methods to determine the temperature by gate-dependent measurements, which cannot be done in the STM. We do not want to speculate at this point, if our method is the most efficient in the case of quantum dots.

The theoretical analysis of the method we just sketched will be published in another form. Experimentally, we need to first test the method by performing calibration and temperature-dependent measurements. We intend to pursue this line of research in the future, but it departs significantly from the main scope of the current manuscript.

We have added a subsection about thermometry in the SI, as an outlook.

Answers to Referee #2:

I reevaluated the manuscript by reading into the paper. Although some interests arose in my second reading, I persist to think that the work is too specialized, not for a general or broad interests. For example, the YSR states have been known for many years, it can help to resolve the pairing symmetry of some new and unknown superconductors. This tip-distance (transmission rate after the author's terming) controlled physics is quite complicated with this novel junction, which can generally give rise to some evolution of the spectrum. I have many concerns about the magnetic moment on the tip apex, the junction itself, and the interpretation of the output results. Thus I still hold the negative attitude to this work.

We thank the Referee again for the assessment of our work. We are happy to see that our response triggered more interest in our manuscript. However, we want to point out that the Referee's opinion our work is too specialized and not of general or broad interests is not shared by us nor the other two Referees.

We first want to mention that in contrast to the Referee's statement, YSR states have indeed gained substantial attention in the last years, mainly triggered by advances in STM-devices that now allow to reveal even minute details of the many-body physics and to control them by external fields. This in turn opened a door to explore a broad range of fundamental questions, in particular, their relevance in the context of topological states, in the context of quantum sensing, and in the context of quantum computing.

More specifically, changing the tip-sample distance is a convenient way to tune the coupling of the impurity to the substrate and hence the energy of the YSR state. It is no more complicated than in other tunnel junctions and already quite well understood as this technique is used by a number of groups as well (e.g. Ref. [7], [11], [16], [18]). Therefore, describing YSR states as "known for many years" and "complicated" at the same time, seems contradictory.

1. Firstly, the authors believe that the magnetic moment on a vanadium adatom on the very apex is spin 1/2, this needs to be well proved. I think this is a very difficult point.

The exact way for how the vanadium atom is attached at the tip is very crucial for the final results. For example, is it side-attached or exactly on the tip? I believe this can give enormous influence on the final results, including the YSR states. I must say that the original proposed YSR states appear at or nearby the magnetic impurity embedded in a body superconductor, it is a collective effect and even some Friedel oscillations of bound states can be observed. Now the magnetic impurity is at the tip, that is really something special, the original picture of YSR states is greatly altered.

We would like to reiterate our response from the previous round, where we have already elaborated on the different pieces of evidence for a spin-1/2 impurity at the tip apex. Over the past few years, we have collected multiple indications for their spin-1/2 character, such as only one YSR state in the gap (i.e. two peaks in the spectrum), Kondo peak splitting beyond a critical magnetic field (see Fig. 4), and transport through two YSR states. A number of these results with impurities at the vanadium tip as well as in the vanadium sample are already published (see Ref. [38], [26], [19]; Huang et al., Phys. Rev. Res. 3, L032007 (2021)), all of which indicate that the impurities in vanadium are spin-1/2 systems giving an overall consistent picture. We would like to point out that the observation of Friedel oscillations is not a necessary condition for observing YSR states. Many YSR states, which can be found in the literature do not exhibit Friedel oscillations and for those, who do exhibit Friedel oscillations, the peak intensity diminishes very quickly moving away from the impurity, i. e. within a few lattice constants at most. Furthermore, since the YSR peak intensity is always very high, we believe that the YSR state is at the tip apex in our case. Furthermore, the YSR states can be nicely described by the Anderson impurity model (already in the mean field approximation), which describes a zero-dimensional impurity attached to a superconducting bulk continuum (see e.g. Ref. [18], [21]). From this point of view, there is no difference between the impurity being attached to the tip apex or the sample surface.

2. A strange configuration is chosen by the authors here. They used a vanadium tip and a vanadium substrate, both are superconductive at zero field. They wrote in the paper that "The sample is already normal conducting at 750 mT, such that there is no shift of the YSR peak by Δ_s . The YSR tip is still superconducting due to the MTF effect." I wonder why the authors don't use a non-superconductive metal as the substrate, like Cu or Au crystal. In that case, the output would be more convincing and easy to explain.

We do not quite understand the Referee's comment here. Since the vanadium sample is normal conducting for the relevant measurements in the magnetic fields, the spectra would look pretty much the same as they do in Fig. 2 (even the analysis in panel d). We use vanadium as sample because it greatly facilitates the preparation of the YSR state at the tip apex. Even more importantly, we can exploit the dimensional confinement in the sharp tip apex to apply much higher magnetic fields before the superconductiv-

ity quenches than in the bulk sample (Meservey-Tedrow-Fulde (MTF) effect (see Ref. [27])). Therefore, we place the YSR state at the tip apex. As the tip has to be superconducting in this case to exploit the MTF effect, accidentally picking up atoms from a non-superconducting sample for whatever reason renders the tip normal conducting due to the proximity effect. We have actually tried working with a superconducting tip on a non-superconducting sample in the past, which was unfortunately unsuccessful due to exactly this problem. So, we kindly ask the referee to also consider the practical point of view to carry out our experiments using a vanadium sample, which we believe to be just as convincing.

Answers to Referee #3:

I thank the authors for considering my suggestions and for their explanations. I stay with my opinion that this is sound work which brings advances for the field. Therefore the results should be published.

We thank the referee for approving publication.